# Association of Insulin Resistance with Vascular Ageing in a General Caucasian Population: An EVA Study

**DOI:** 10.3390/jcm10245748

**Published:** 2021-12-08

**Authors:** Leticia Gómez-Sánchez, Marta Gómez-Sánchez, Cristina Lugones-Sánchez, Olaya Tamayo-Morales, Susana González-Sánchez, Emiliano Rodríguez-Sánchez, Luis García-Ortiz, Manuel A. Gómez-Marcos

**Affiliations:** 1Institute of Biomedical Research of Salamanca (IBSAL), 37007 Salamanca, Spain; leticiagmzsnchz@gmail.com (L.G.-S.); martagmzsnchz@gmail.com (M.G.-S.); cristinals@usal.es (C.L.-S.); olayatm@usal.es (O.T.-M.); gongar04@gmail.com (S.G.-S.); emiliano@usal.es (E.R.-S.); lgarciao@usal.es (L.G.-O.); 2Primary Care Research Unit of Salamanca (APISAL), 37005 Salamanca, Spain; 3Health Service of Castile and Leon (SACyL), 37007 Salamanca, Spain; 4Faculty of Medicine, University of Salamanca, 37007 Salamanca, Spain

**Keywords:** insulin resistance, vascular ageing, arterial stiffness

## Abstract

The data on the relationship between insulin resistance and vascular ageing are limited. The aim of this study was to explore the association of different indices of insulin resistance with vascular ageing in an adult Caucasian population without cardiovascular disease. We selected 501 individuals without cardiovascular disease (mean age: 55.9 years, 50.3% women) through random sampling stratified by sex and age. Arterial stiffness was evaluated by measuring the carotid-to-femoral pulse wave velocity (cfPWV) and brachial-to-ankle pulse wave velocity (baPWV). The participants were classified into three groups according to the degree of vascular ageing: early vascular ageing (EVA), normal vascular ageing (NVA) and healthy vascular ageing (HVA). Insulin resistance was evaluated with the homeostatic model assessment of insulin resistance (HOMA-IR) and another five indices. The prevalence of HVA and EVA was 8.4% and 21.4%, respectively, when using cfPWV, and 7.4% and 19.2%, respectively, when using baPWV. The deterioration of vascular ageing, with both measurements, presented as an increase in all the analysed indices of insulin resistance. In the multiple regression analysis and logistic regression analysis, the indices of insulin resistance showed a positive association with cfPWV and baPWV and with EVA.

## 1. Introduction

Insulin resistance produces hyperglycaemia and dyslipidaemia, and is characterised by high levels of plasma triglycerides and low levels of high-density lipoproteins (HDLs) [1,2]. All this, together with abdominal obesity [3], leads to endothelial dysfunction [4] and increases arterial stiffness [5,6,7], atherosclerosis [2,8] and morbimortality from cardiovascular diseases [1,2,9,10,11]. The homeostatic model assessment of insulin resistance (HOMA-IR) is the method most commonly used to evaluate insulin resistance in clinical practice [12]. However, this method is expensive [13]. Therefore, based on anthropometric, glycaemic and/or biochemical parameters gathered routinely in clinical practice, new indices have been created to evaluate insulin resistance, such as the triglyceride and glucose index (TyG index) [8,14,15], triglyceride-to-high-density lipoprotein cholesterol ratio (T/HDL-c ratio) [16], lipid accumulation product index (LAP index) [17,18], waist-to-height ratio (WHt ratio) and visceral adiposity index (VA index) [19]. These indices have shown good correlation with HOMA-IR [14,17,19,20] and an independent association between insulin resistance and cardiovascular events [21]; thus, they can be used as substitute measures to assess insulin resistance [5,22,23].

Furthermore, increases in central arterial stiffness, evaluated with carotid-to-femoral pulse wave velocity (cfPWV) [24,25], and peripheral arterial stiffness, evaluated with brachial ankle pulse wave velocity (baPWV) [26], can be used to predict morbimortality by cardiovascular diseases. It is also known that increases in insulin resistance increase arterial stiffness [27]. Vascular ageing is influenced by arterial stiffness, reflecting a dissociation between the chronological age and biological age of the main arteries, with such alterations preceding the appearance of cardiovascular events [28,29,30]. Numerous epidemiological studies have been carried out to determine the influencing factors of vascular ageing, which have attracted considerable interest because they show a stronger relationship with morbimortality from cardiovascular diseases than biological ageing [28,30]. However, few studies have analysed the relationship between vascular ageing and insulin resistance [31,32].

Therefore, it is of increasing research interest to determine the interactions that link insulin resistance to vascular ageing, as well as to evaluate which insulin resistance index is the most adequate to predict vascular ageing. This relationship has not been explored in the Caucasian population; thus, the main objective of the present study was to analyse the association between different insulin resistance indices and vascular ageing in an adult Caucasian population without cardiovascular disease. As a secondary objective, we analysed whether this association differs depending on the arterial stiffness measurements used to define vascular ageing.

## 2. Materials and Methods

### 2.1. Study Population

This was a cross-sectional descriptive study with individuals recruited from the association between different risk factors and vascular accelerated ageing study (EVA study) (NCT02623894) [33].

The included individuals were from an urban population of 43,946 people. Through random sampling with replacements stratified by sex and age group (35, 45, 55, 65 and 75 years), 501 Spanish individuals were selected, with approximately 100 in each of the groups (50 women and 50 men). The recruitment was carried out between June 2016 and November 2017. Inclusion criteria: patients aged 35–75 years who agreed to participate in the study and did not meet any of the exclusion criteria. Exclusion criteria: participants who were in terminal condition, could not travel to the health centres to undergo the corresponding examinations, or did not wish to sign the informed consent; participants with a medical history of CVD (ischaemic heart disease or stroke, peripheral arterial disease or heart failure), a diagnosis of renal failure in terminal stages (glomerular filtration rate below 30%), chronic inflammatory disease or acute inflammatory process in the past three months; and patients treated with oestrogens, testosterone or growth hormones. A detailed description of the reference population, as well as the inclusion and exclusion criteria and the causes by age group and sex, is presented in Figure 1.

Before the participants were included in the study, we informed them about its content, and they all signed informed consent. The study was approved on 4 May 2015 by the Drug Research Ethics Committee of Salamanca. Throughout the course of the study, the recommendations of the Declaration of Helsinki were followed [34].

### 2.2. Variables and Measurement Instruments

A detailed description of the study procedures, as well as the inclusion and exclusion criteria and the response rate, have previously been published [33,35]. Before initiating the study, two healthcare professionals were trained to record the measurements and gather the necessary questionnaires, following a standardised protocol.

### 2.3. Measurement of Arterial Stiffness

cfPWV was measured in the supine position, using a SphygmoCor^®^ device (AtCor Medical Pty Ltd., Head Office, West Ryde, Australia) [36]. The carotid and femoral pulse waves were recorded, estimating the time of delay in comparison with the ECG wave and calculating the pulse wave velocity. The distances were measured using a measuring tape from the sternal notch to the point where the sensor was placed over the carotid and femoral arteries [36]. The pulse wave was evaluated by flattening tonometry, using a sensor over the radial artery; through a mathematical transformation, the device estimated the aortic pulse wave.

baPWV was measured using a VaSera VS-1500® device (Fukuda Denshi, Denshi Co. Ltd., Tokyo, Japan)), which employs an oscillometer method, following the manufacturer’s instructions. The participants were asked not to smoke or consume caffeine 1 h before the test; they wore comfortable clothing and rested for at least 10 min before the test. The cuffs were adapted to the circumference of the arms and ankles in both the left and right limbs. baPWV was estimated using the following equation: baPWV = (0.5934 × height (cm) + 14.4724)/TBA, where TBA is the time interval between the waves of the arm and ankle [37].

### 2.4. Definition of Vascular Ageing

Vascular ageing was defined in three steps:Step 1: the participants who presented with vascular injury in the carotid artery or peripheral arteriopathy, using the criteria established in the 2018 clinical practice guidelines of the European Societies of Hypertension and Cardiology for the treatment of arterial hypertension [38], were classified as early vascular ageing (EVA);Step 2: with the percentiles of arterial stiffness, we used two criteria: the 10th and 90th percentiles of cfPWV and the 10th and 90th percentiles of baPWV of the population studied by age and sex. The individuals with values of cfPWV or baPWV over p90 were considered EVA; those between p10 and p90 were classified as normal vascular ageing (NVA); and those with values below p10 were classified as healthy vascular ageing (HVA);Step 3: the individuals diagnosed with type 2 diabetes mellitus or hypertension were included in the HVA group; those with the criteria in the percentiles for cfPWV or those of baPWV were classified as NVA [39]. The distribution of participants fitting the two criteria in each of the groups is shown in Figure 2. The percentiles by age groups and sex of cfPWV and baPWV are shown in Appendix A: Appendix A.

### 2.5. Insulin Resistance Indices

The homeostatic model assessment of insulin resistance (HOMA-IR) was estimated using the following formula: (fasting glucose in mmol/L) × (fasting insulin in μU/mL)/ 22.5 [12]. The triglyceride-to-high-density lipoprotein cholesterol ratio (T/HDL-C ratio) was calculated using the following equation: (triglyceride in mg/dL)/(high-density lipoprotein cholesterol in mg/dL) [16]. The triglyceride and glucose index (TyG index) was calculated using the following formula: Ln [(fasting triglyceride in mg/dL × fasting glucose in mg/dL)/2] [14]. The lipid accumulation product index (LAP index) was calculated using the following formula: [(waist circumference (WC) in cm − 65) × (triglyceride in mmol/L)] in men and [(WC in cm − 58) × (triglyceride in mmol/L)] in women [17]. The waist-to-height ratio (WHt ratio) was calculated using the following formula: WHt ratio = (WC in cm/height in cm). The visceral adiposity index (VA index) [19] was calculated using the following formula: VA index = (WC/(39.68 + 1.88 × BMI)) × (triglyceride/1.03) × (1.31/LDL-c) in men and VA index = (WC/(36.58 + 1.89 × BMI)) × (triglyceride/0.81) × (1.52/LDL-c) in women, measuring triglycerides and LDL-c mml/L [19].

### 2.6. Measurement of Vascular Injury

The carotid intima–media thickness was measured using a Sonosite Micromaxx ultrasound device (Sonosite Inc., Bothell, Washington, DC, USA), with Sonocal software. The common carotid artery was measured after examining a 10 mm longitudinal section at a distance of 1 cm from the fork, and measurements of the proximal and distal walls were taken. The presence of peripheral artery disease was assessed by calculating the ankle–brachial index using VaSera VS-1500 (Fukuda Denshi, Denshi Co. Ltd., Tokyo, Japan), following the criteria of the 2018 ESC/ESH Guidelines for diagnosing and treating hypertension in order to establish the presence of vascular injury [29].

### 2.7. Laboratory Analyses

Blood samples were collected at the Primary Healthcare Research Unit between 8 am and 9 am, with the participants fasting for at least 12 h beforehand, using a standardised venepuncture protocol; the blood samples were processed in the laboratory of the reference hospital. Glucose, triglycerides, total cholesterol and HDL-c were analysed using enzymatic methods. LDL-c was estimated with the Friedewald equation. Fasting insulin was analysed through immunoassay methods.

We considered that the individuals presented normal glucose metabolism if fasting glycemia *<* 100 mg/dL, HbA1c *<* 5.7% and no antidiabetic drugs were being administered. Prediabetes was considered if fasting glycemia values were between 100 and 125 mg/dL, or HbA1c was between 5.7% and 6.4% and no antidiabetic drugs were being administered. Type 2 diabetes mellitus was considered if fasting glycemia ≥ 126 mg/dL, HBa1c ≥ 6.5%, or the participant was receiving antidiabetic drugs.

The rest of the variables used in this study were analysed, as reflected in the protocol [33].

### 2.8. Statistical Analysis

The data of the continuous variables are shown as mean ± standard deviation, and those of the categorical variables are numbers and percentages. Comparison of means between two independent groups was performed with Student’s t-tests; for more than two groups, a one-way analysis of variance (ANOVA) was used. A post hoc analysis to assess the differences between more than two groups was performed with the least significant difference (LSD) test. In the comparison between categorical variables, the χ^2^ test was used. Correlations between cfPWV and baPWV with the indices of insulin resistance were assayed with Pearson’s correlation coefficient. The association between the insulin resistance indices and the stiffness measurements used was evaluated through a multiple regression analysis. To analyse the association of the insulin resistance indices with vascular ageing, a logistic regression model was developed for each of the studied indices. We used HVA, NVA and EVA as dependent variables. In model 1, we coded HVA and NVA = 0 and EVA = 1. In model 2, we coded HVA = 0, and NVA and EVA = 1. Lastly, in model 3, we coded HVA = 0 and EVA = 1. We used the six insulin resistance indices analysed as independent variables. The three models were repeated with the two classifications of vascular ageing conducted using cfPWV and baPWV. In all multiple and logistic regression models, the adjustment variables were age, sex (0 = woman, 1 = man) and the intake of antihypertensive, lipid-lowering and hypoglycaemic drugs (no drug intake = 0, and drug intake = 1). In the hypothesis test, an α risk of 0.05 was established as the limit of statistical significance. All analyses were carried out using SPSS statistical software for Windows, v.25.0 (IBM Corp, Armonk, NY, USA).

## 3. Results

### 3.1. Baseline Characteristics

The clinical measures, cardiovascular risk factors, arterial stiffness measures, drug intake and parameters to measure insulin resistance, globally and by sex, are described in Table 1. The mean age was 55.90 ± 14.24 years. The insulin resistance indices were greater in men, except for HOMA-IR and the VA index.

Using cfPWV, the global prevalence of HVA was 8.4% (8.0% in men and 8.7% in women), and that of EVA was 21.4% (25.7% in men and 17.1% in women). Using baPWV, the global prevalence of HVA was 7.4% (6.4% in men and 8.4% in women), and that of EVA was 19.2% (22.9% in men and 15.5% in women) (Figure 3).

### 3.2. Relationship of the Insulin Resistance Indices with Vascular Ageing

Table 2 shows the mean values of the insulin resistance indices analysed in individuals with HVA, NVA and EVA, using the percentiles of cfPWV as a measurement of arterial stiffness to classify the individuals according to the degree of vascular ageing. Globally and in women, the deterioration of vascular ageing presented with an increase in all the insulin resistance indices. In men, the increases in the T/HDL-c ratio and VA index were not significant (*p* > 0.05).

Table 3 shows the mean values of the insulin resistance indices analysed in individuals with HVA, NVA and EVA, using the percentiles of baPWV as a measurement of arterial stiffness to classify the individuals according to the degree of vascular ageing. Globally, vascular deterioration presented with an increase in all the insulin resistance indices. In men, increases in the T/HDL-c ratio, LAP index and VA index, as well as those in the WHt ratio and LAP index in women, were not significant (*p* > 0.05).

Correlations between the measures of arterial stiffness and the insulin resistance indices are shown in Appendix A: Appendix A; there was a positive correlation in all cases, except between cfPWV and the T/HDL-c ratio in men. The correlation persisted in all cases after controlling for age and sex, as can be observed in Appendix A: Appendix A.

The values of the two measurements of arterial stiffness in both sexes, according to impaired glucose metabolism, are shown in Appendix A: Appendix A.

### 3.3. Association of the Insulin Resistance Indices with Arterial Stiffness and Vascular Ageing

In the multiple regression analysis, all the insulin resistance indices showed a positive association with the two stiffness measurements analysed, after controlling for possible confounders (Table 4).

In model 1 of the logistic regression analysis, there were high OR values with GyT index (OR = 1.01 and OR = 1.02), T/HDL-c ratio (OR = 1.15 and OR = 1.21), LAP index (OR = 1.02 and OR = 1.01) and VA index (OR = 1.09 and OR = 1.12) using cfPWV and baPWV as measurements of arterial stiffness in individuals with EVA with respect to the remaining participants (Figure 4).

The results obtained from the logistic regression analysis using model 2 and model 3 can be seen in Appendix A: Appendix A and Appendix A.

**Figure 4 jcm-10-05748-f004:**
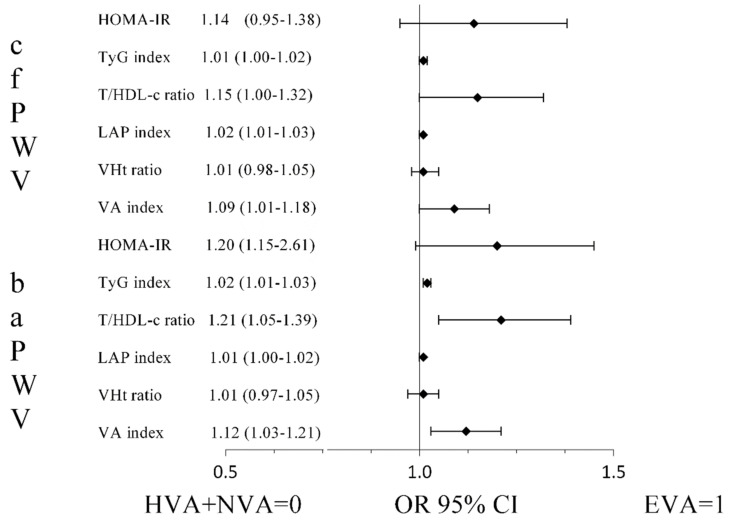
Odds ratio of determinants of EVA vs. HVA + NVA with insulin resistance indexes in adults aged 35–75 years. Adjusted for age, sex, antihypertensive, lipid-lowering and hypoglycaemic drugs. Abbreviations: EVA, early vascular ageing; HVA, healthy vascular ageing; NVA, normal vascular ageing; cfPWV, carotid-to-femoral pulse wave velocity; baPWV, brachial-to-ankle pulse wave velocity; HOMA-IR, homeostatic model assessment of insulin resistance; TyG index, triglyceride and glucose index; T/HDL-c ratio, triglyceride-to-high-density lipoprotein cholesterol ratio; LAP index, lipid accumulation product index; WHt ratio, waist-to-height ratio; VA index, visceral adiposity index.

## 4. Discussion

This study analysed the association between different insulin resistance indices and vascular ageing and arterial stiffness measurements using cfPWV and baPWV in an adult Caucasian population without cardiovascular disease. Regarding the main findings of this study, the insulin resistance indices showed a positive association with vascular ageing, although there were differences between them. All the insulin resistance indices analysed were related to central and peripheral arterial stiffness.

The prevalence of HVA and EVA has been analysed in previous studies [32,40,41,42,43], all of which agree with the results of the present study, indicating that the prevalence of EVA is greater in men, and that of HVA is greater in women. However, the prevalence rates found between the studies are different and cannot be compared, because the cut-off points used, the mean age and the distribution by sex, as well as the prevalence of the different cardiovascular risk factors and race differ from one study to another.

The positive correlations found in this study between the arterial stiffness measurements (cfPWV and baPWV for the evaluation of central and peripheral arterial stiffness, respectively) and the different indices that reflect peripheral or hepatic insulin resistance are in line with those reported in previous studies. For example, the recent study of Poon et al. [5], conducted in US adults without diabetes, found that HOMA-IR, the T/HDL-c ratio and the TyG index were positively related to central arterial stiffness. This association between central arterial stiffness and insulin resistance has also been found in Chinese adults without diabetes mellitus [44] or with arterial hypertension [45]. In Korean adults, HOMA-IR and the TyG index have been shown to exhibit a positive association with peripheral arterial stiffness [21,46]. Lastly, the results of the study entitled “elderly individuals: the Northern Shanghai Study” [20] demonstrated an association between the TyG index and central and peripheral arterial stiffness. However, this study complements the existing information, because it analysed the association of central and peripheral arterial stiffness evaluated with cfPWV and baPWV with the indices of peripheral and hepatic insulin resistance in the same sample of individuals without previous cardiovascular disease. Therefore, we have confirmed a relationship between insulin resistance and central and peripheral arterial stiffness in an adult population, using indices that reflect more than one aspect of insulin resistance.

The participants characterised by HVA had lower values for the insulin resistance indices than the participants with EVA, which is in line with the results published in the study of Framingham [32]. In this study, the participants with HVA had lower values of HOMA-IR than the participants without HVA. However, most of the published studies did not analyse the association of the different insulin resistance indices with vascular ageing, focusing on the relationship of the parameters of abdominal obesity, lipid components and the glycemia values used to calculate the different indices of insulin resistance. Thus, numerous studies have shown that abdominal obesity is positively associated with vascular ageing [28,30,32,40,41,47,48,49]. The increase in triglycerides and the decrease in HDL-c show a correlation with vascular ageing [30,32,40,41,42]. In this sense, the association between glycometabolic impairment and an increase in arterial stiffness is consistent with previous studies [50], and a better control of glycemia alleviates or prevents the progression of arterial stiffness in individuals with type 2 diabetes mellitus [51]. Similarly, it has been demonstrated that glycometabolic impairment is associated with vascular ageing. The study by Framingham [32] revealed that the absence of type 2 diabetes mellitus was associated with the presence of HVA, and that the individuals with type 2 diabetes mellitus were eight times less likely to present with HVA compared to non-diabetic individuals; in the study of Shanghai [41] and in the MARE study [40], a baseline glycemia level over 100 mg/dL was associated with EVA. Lastly, in a study carried out in Chinese adults, it was found that higher fasting glucose levels and HbA1c were associated with EVA [52].

Furthermore, the different behaviours of the insulin resistance indices with vascular ageing are in agreement with the results of previous studies, which showed that the TyG index was more strongly related to greater arterial stiffness than HOMA-IR in Korean adults [21].

In summary, this study provides further support to the available information which indicates that insulin resistance is an independent predictor of vascular ageing in the general population.

## 5. Conclusions

In general, all the insulin resistance indices showed a positive association with vascular ageing and with central and peripheral arterial stiffness. However, we have found differences in the associations between different indices of insulin resistance and ageing. Therefore, prospective studies are necessary to clarify these results.

## 6. Limits

Regarding the main limitations of this study, it is important to highlight that, firstly, the cross-sectional study design does not allow for inferring causality. Secondly, the results of this study only refer to a Caucasian urban population, and thus cannot be generalised to other races/ethnicities. Lastly, the prevalence of cardiovascular risk factors in this study was lower than in other studies conducted in a Caucasian population. The main strengths of this study were the recruitment of participants by population-based random sampling, and the fact that this is the first study to analyse the relationship between vascular ageing and different insulin resistance indices.

## Figures and Tables

**Figure 1 jcm-10-05748-f001:**
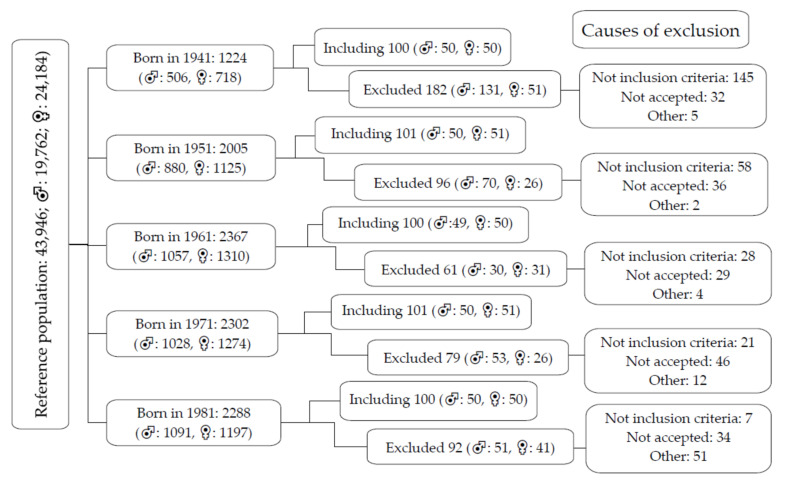
Flow diagram describing the reference population (43,946), those included and excluded, as well as exclusion criteria by age group and sex. A total of 259 subjects did not meet the inclusion criteria; 177 did not agree to participate in the study; 74 (other) subjects were not located due to changes in address or telephone number.

**Figure 2 jcm-10-05748-f002:**
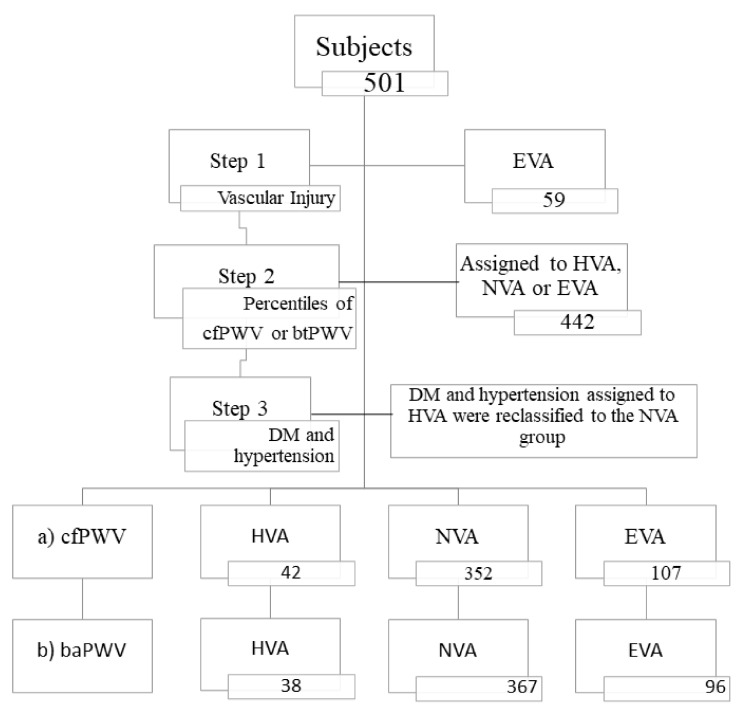
Distribution of participants fitting the two criteria in each of the groups: healthy vascular ageing, normal vascular ageing, and early vascular ageing. (**a**) The 10th and 90th carotid-to-femoral pulse wave velocity percentiles of the population studied by age and sex; (**b**) the 10th and 90th brachial-to-ankle pulse wave velocity percentiles of the population studied by age and sex, classified thus: above the 90th percentile was considered early vascular ageing; between the 10th and 90th percentiles was considered normal vascular ageing; and below the 10th percentile was considered healthy vascular ageing. Abbreviations: EVA, early vascular ageing; HVA, healthy vascular ageing; NVA, normal vascular ageing; cfPWV, carotid-to-femoral pulse wave velocity, baPWV brachial-to-ankle pulse wave velocity; DM, diabetes mellitus.

**Figure 3 jcm-10-05748-f003:**
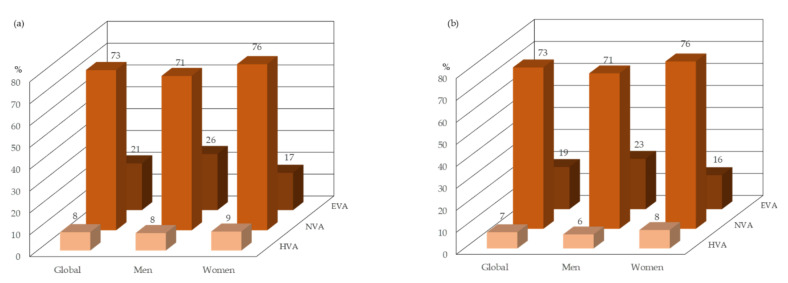
Proportion of vascular ageing status global and by sex. (**a**) cfPWV and (**b**) baPWV. Abbreviations: EVA, early vascular ageing; HVA, healthy vascular ageing; NVA, normal vascular ageing; cfPWV, carotid-to-femoral pulse wave velocity; baPWV, brachial-to-ankle pulse wave velocity.

**Table 1 jcm-10-05748-t001:** General characteristics of the subjects, globally and by sex.

	Global (501)	Men (248)	Women (251)	*p* Value
Age, years	55.90 ± 14.24	55.95 ± 14.31	55.85 ± 14.19	0.934
Clinical measures, mean (SD)				
SBP, mmHg	121 ± 23	126 ± 20	115 ± 25	<0.001
DBP, mmHg	76 ± 10	77 ± 9	74 ± 10	<0.001
Total cholesterol, mg/dL	195 ± 32	191 ± 32	197 ± 31	0.142
LDL cholesterol, mg/dL	116 ± 29	117 ± 30	114 ± 29	0.148
HDL cholesterol, mg/dL	59 ± 16	53 ± 14	64 ± 16	<0.001
Triglyceride, mg/dL	103 ± 53	112 ± 54	94 ± 50	<0.001
Fasting glucose, mg/dL	88 ± 17	90 ± 19	86 ± 16	0.013
HbA1c, (%)	5.49 ± 0.56	5.54 ± 0.63	5.44 ± 0.47	<0.001
Fasting insulin, μU/mL	8.45 ± 4.66	8.78 ± 4.59	8.11 ± 4.71	0.110
Weight, kg	72 ± 14	79 ± 12	66 ± 12	<0.001
Height, cm	165 ± 10	172 ± 7	159 ± 7	<0.001
BMI, kg/m^2^	26.52 ± 4.23	26.90 ± 3.54	26.14 ± 4.79	0.044
Waist circumference, cm	93.33 ± 12.00	98.76 ± 9.65	87.95 ± 11.68	<0.001
Arterial stiffness, mean (SD)				
cfPWV, m/s	8.17 ± 2.53	8.58 ± 2.74	7.77 ± 2.24	0.043
baPWV, m/s	12.93 ± 2.68	13.16 ± 2.46	13.16 ± 2.46	0.064
Cardiovascular Risk factors, n (%)				
Smoker	90 (17.96)	49 (9.80)	41 8.20)	0.353
Hypertension	147 (29.34)	82 (32.93)	65 (25.79)	0.095
Dyslipidaemia	191 (38.12)	95 (38.10)	96 (38.20)	0.989
Normal glucose	358 (71.42)	174 (69.90)	184 (73.00)	0.314
Prediabetes	105 (21.00)	49 (19.70)	56 (22.20)	0.459
Diabetes mellitus	38 (7.58)	26 (10.40)	12 (4.80)	0.018
Obesity	94 (18.76)	42 (16.90)	52 (20.60)	0.304
Abdominal obesity	193 (38.52)	78 (31.30)	115 (45.80)	0.001
Medication use, n (%)				
Antihypertensive drugs	147 (29)	82 (33)	65 (26)	0.095
Lipid-lowering drugs	102 (20)	49 (20)	53 (21)	0.740
Hypoglycaemic drugs	35 (7)	23 (9)	12 (5)	0.055
Insulin resistance indexes, mean (SD)				
HOMA-IR	1.87 ± 1.18	1.98 ± 1.18	1.77 ± 1.17	0.052
TyG index	10.12 ± 1.15	10.37 ± 1.06	9.86 ± 1.18	<0.001
T/HDL-c ratio	2.01 ± 1.49	2.36 ± 1.59	1.66 ± 1.29	<0.001
LAP index	43.10 ± 31.08	48.68 ± 30.51	37.39 ± 30.67	<0.001
WHt ratio	0.57 ± 0.07	0.58 ± 0.06	0.56 ± 0.08	0.001
VA index	3.27 ± 2.44	3.31 ± 2.26	3.21 ± 2.61	0.656

SBP, systolic blood pressure; DBP, diastolic blood pressure; LDL, low-density lipoprotein; HDL, high-density lipoprotein; BMI, body mass index; cfPWV, carotid-to-femoral aortic pulse wave velocity; baPWV, brachial-to-ankle aortic pulse wave velocity; HOMA-IR, homeostatic model assessment of insulin resistance; HbA1c, glycosylated haemoglobin; TyG index, triglyceride and glucose index; T/HDL-c ratio, triglyceride-to-high-density lipoprotein cholesterol ratio; LAP index, lipid accumulation product index; WHt ratio, waist-to-height ratio; VA index, visceral adiposity index.

**Table 2 jcm-10-05748-t002:** Insulin resistance indexes to the degree of vascular ageing using the 10th percentile and 90th percentile of cfPWV.

	HVA (42)	NVA (253)	EVA (107)	*p* Value
Global, mean (SD)				
HOMA-IR, *^,¥,&^	1.36 ± 0.64	1.85 ± 1.14	2.17 ± 1.37	0.001
TyG index, *^,¥,&^	179 ± 30	197 ± 40	220 ± 70	<0.001
T/HDL-c ratio, ^¥,&^	1.54 ± 0.99	1.95 ± 1.43	2.38 ± 1.73	0.003
LAP index, *^,¥,&^	26.26 ± 15.54	42.52 ± 31.20	51.62 ± 32.37	<0.001
WHt ratio, *^,¥,&^	0.53 ± 0.06	0.57 ± 0.07	0.58 ± 0.08	0.019
VA index, *^,¥,&^	2.39 ± 1.31	3.20 ± 2.40	3.85 ± 2.79	0.004
Men, mean (SD)				
HOMA-IR, *^,¥^	1.39 ± 0.80	1.98 ± 1.22	2.16 ± 1.13	0.038
TyG, mg/dL, *^,¥,&^	187 ± 28	204 ± 38	228 ± 73	0.001
T/HDL-c ratio	2.05 ± 1.18	2.28 ± 1.56	2.67 ± 1.74	0.165
LAP index, *^,¥^	32.25 ± 15.43	47.73 ± 30.40	56.28 ± 32.25	0.007
WHt ratio, *^,¥^	0.54 ± 0.04	0.58 ± 0.06	0.59 ± 0.06	0.005
VA index	2.77 ± 1.65	3.18 ± 2.22	3.86 ± 2.46	0.067
Women, mean (SD)				
HOMA-IR, ^¥,&^	1.33 ± 0.44	1.73 ± 1.05	2.18 ± 1.68	0.015
TyG index, *^, ¥,&^	172 ± 32	191 ± 41	209 ± 64	0.006
T/HDL-c ratio, *^,¥^	1.07 ± 0.43	1.67 ± 1.25	1.94 ± 1.63	0.033
LAP index, *^,&^	20.83 ± 13.81	37.93 ± 31.25	44.69 ± 31.64	0.011
WHt ratio, *^,¥^	0.51 ± 0.07	0.56 ± 0.08	0.58 ± 0.09	0.012
VA index, *^,&^	2.06 ± 0.82	3.21 ± 2.55	3.80 ± 3.23	0.039

HVA, healthy vascular ageing; NVA, normal vascular ageing; EVA, early vascular ageing; cfPWV, carotid-to-femoral aortic pulse wave velocity; HOMA-IR, homeostatic model assessment of insulin resistance; TyG index, triglyceride and glucose index; T/HDL-c ratio, triglyceride-to-high-density lipoprotein cholesterol ratio; LAP index, lipid accumulation product index; WHt ratio, waist-to-height ratio; VA index, visceral adiposity index. Differences among groups: continuous variables analysis of variance and post hoc using the least significant difference tests. * *p* value less than 0.05 between HVA and NVA. ^¥^
*p* value less than 0.05 between NVA and EVA. ^&^
*p* value less than 0.05 between HVA and EVA.

**Table 3 jcm-10-05748-t003:** Insulin resistance indexes to the degree of vascular ageing using the 10th percentile and 90th percentile of baPWV.

	HVA (38)	NVA (367)	EVA (96)	*p* Value
Global, mean (SD)				
HOMA-IR, *^,¥,&^	1.23 ± 0.61	1.85 ± 1.13	2.20 ± 1.38	<0.001
TyG index, *^,¥,&^	180 ± 28	198 ± 42	219 ± 68	<0.001
T/HDL-c ratio, ^¥,&^	1.63 ± 1.16	1.93 ± 1.39	2.47 ± 1.84	0.002
LAP index, *^,&^	34.30 ± 23.60	41.66 ± 31.01	51.76 ± 32.27	0.004
WHt ratio, ^¥,&^	0.54 ± 0.06	0.56 ± 0.07	0.58 ± 0.07	0.019
VA index, ^¥,&^	2.63 ± 1.80	3.15 ± 2.31	3.95 ± 2.97	0.005
Men, mean (SD)				
HOMA-IR, *^,¥^	1.15 ± 0.59	1.98 ± 1.20	2.20 ± 1.15	0.007
TyG index, *^,¥,&^	192 ± 23	204 ± 39	228 ± 74	0.002
T/HDL-c ratio	2.13 ± 1.20	2.27 ± 1.52	2.69 ± 1.86	0.186
LAP index	38.95 ± 21.16	47.12 ± 30.53	56.23 ± 31.57	0.061
WHt ratio, *^,¥^	0.54 ± 0.05	0.58 ± 0.06	0.59 ± 0.06	0.010
VA index	2.96 ± 1.81	3.19 ± 2.16	3.82 ± 2.63	0.158
Women, mean (SD)				
HOMA-IR, ^¥,&^	1.29 ± 0.64	1.73 ± 1.06	2.19 ± 1.68	0.014
TyG index, *^,¥,&^	171 ± 29	192 ± 44	206 ± 56	0.017
T/HDL-c ratio, ^¥,&^	1.25 ± 0.99	1.61 ± 1.18	2.14 ± 1.77	0.019
LAP index	30.76 ± 25.22	36.63 ± 30.68	45.23 ± 32.57	0.162
WHt ratio	0.55 ± 0.07	0.55 ± 0.08	0.56 ± 0.09	0.685
VA index, ^¥,&^	2.39 ± 1.80	3.11 ± 2.45	4.13 ± 3.43	0.027

HVA, healthy vascular ageing; NVA, normal vascular ageing; EVA, early vascular ageing; baPWV, brachial-to-ankle aortic pulse wave velocity; HOMA-IR, homeostatic model assessment of insulin resistance; T/HDL-c ratio, triglyceride-to-high-density lipoprotein cholesterol ratio; LAP index, lipid accumulation product index; WHt ratio, waist-to-height ratio; VA index, visceral adiposity index. Differences among groups: continuous variables analysis of variance and post hoc using the least significant difference tests. * *p* value less than 0.05 between HVA and NVA. ^¥^
*p* value less than 0.05 between NVA and EVA. ^&^
*p* value less than 0.05 between HVA and EVA.

**Table 4 jcm-10-05748-t004:** Association of arterial stiffness parameters with insulin resistance indices.

	β	95% IC	*p*
cfPWV, m/sec			
HOMA-IR	0.196	(0.049–0.344)	0.009
TyG index	0.012	(0.004–0.020)	0.004
T/HDL-c ratio	0.137	(0.026–0.248)	0.015
LAP index	0.011	(0.005–0.016)	<0.001
WHt ratio	3.798	(1.293–6.304)	0.003
VA index	0.091	(0.024–0.158)	0.008
baPWV, m/sec			
HOMA-IR	0.203	(0.060–0.346)	0.005
TyG index	0.007	(0.003–0.011)	0.002
T/HDL-c ratio	0.227	(0.118–0.335)	<0.001
LAP index	0.009	(0.004–0.014)	0.001
WHt ratio	2.848	(0.365–5.331)	0.025
VA index	0.134	(0.069–0.199)	<0.001

cfPWV, carotid-to-femoral aortic pulse wave velocity; baPWV, brachial-to-ankle aortic pulse wave velocity; HOMA-IR, homeostatic model assessment of insulin resistance; TyG index, triglyceride and glucose index; T/HDL-c ratio, triglyceride-to-high-density lipoprotein cholesterol ratio; LAP index, lipid accumulation product index; WHt ratio, waist-to-height ratio; VA index, visceral adiposity index. Multiple regression analysis using cfPWV m/sec and baPWV m/sec as dependent variables, insulin resistance indices (HOMA-IR, HbA1c, TyG, TG/HDL and WHt ratio) as independent variables, and age, sex, hypotensive, hypoglycaemic, and lipid-lowering drugs as adjustment variables.

## Data Availability

The datasets used and/or analysed during the current study are available from the corresponding author on reasonable request.

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
