# Peer review of "Association of Insulin Resistance with Vascular Ageing in a General Caucasian Population: An EVA Study"

_jcm, 2021, doi:10.3390/jcm10245748_

Round 1
Reviewer 1 Report
I really appreciate the work which is well designed and well exposed. I would just like to point out a few things:
- Figure 2 is not very clear and creates a bit of confusion. Why for people born in 1941 for example (and for all other groups), of 1224 individuals 100 are included and 182 excluded (and the others? The graph doesn't clarify the selection)
- Line 124: annd - typo
- Line 283 - ...criteria are not the same....what changes?
- I would like to revise, if possible, the supplementary material, not attached.
Author Response
Comments and Suggestions for Authors
I really appreciate the work which is well designed and well exposed. I would just like to point out a few things:
- Figure 1 is not very clear and creates a bit of confusion. Why for people born in 1941 for example (and for all other groups), of 1224 individuals 100 are included and 182 excluded (and the others? The graph doesn't clarify the selection)
Authors' Answer
The population base of the present study was the list of subjects receiving health care in five urban Spanish primary care health centres (43,946 persons). From this population all subjects born in 1941 (M: 506, F: 718), 1951 (M: 880, F: 1125), 1961 (M: 1057, F: 1310), 1971 (M: 1028, F: 1274) and 1981 (M: 1091, F: 1197) were listed, separated by sex.
From these lists, 50 women and 50 men in each group were selected by random sampling with replacement, a total of 500 subjects. A total of 510 subjects were discarded for the following reasons: 259 subjects did not meet the inclusion criteria, 177 did not agree to participate in the study, and 74 subjects could not be located due to a change of address or telephone number (other). Therefore, these 510 subjects, using random sampling with replacement, were replaced in the respective sampling strata.
In order to clarify the information, we have completed the caption to figure 1. In the new version of the manuscript, it reads as follows:
Figure 1. Flow diagram describing the reference population (43,946), those included and excluded, as well as exclusion criteria by age group and sex. A total of 259 subjects did not meet the inclusion criteria; 177 did not agree to participate in the study; 74 (other) subjects were not located due to changes in address or telephone number.
- Line 124: annd – typo
Authors' Answer
We have corrected the typographical error in line 124 to read as follows in the current version.
The percentiles by age groups and sex of cfPWV and baPWV are shown in Additional file 1: Fig. S1.
- Line 283 - ...criteria are not the same....what changes?
Authors' Answer
The subjects analysed in this study ranged in age from 35 to 75 years with a mean age of 55.90 ± 14.24 years without previous cardiovascular disease, 50% of each sex.
The criteria used to classify arterial ageing in this study were as follows:
- Step 1: the participants who presented vascular injury in the carotid artery or peripheral arteriopathy, using the criteria established in the 2018 clinical practice guidelines of the European Societies of Hypertension and Cardiology for the treatment of arterial hypertension [1], were classified as early vascular aging (EVA).
- Step 2: with the percentiles of arterial stiffness, we used two criteria, which were the 10th and 90th percentiles of cfPWV and the 10th and 90th percentiles of baPWV of the population studied by age and sex. The individuals with values of cfPWV or baPWV over p90 were considered EVA, those between p10 and p90 were classified as normal vascular aging (NVA) and those with values below p10 were classified as healthy vascular aging (HVA).
- Step 3: the individuals diagnosed with type diabetes mellitus or hypertension included in the HVA group, with the criteria of the percentiles of cfPWV or those of baPWV, were classified as NVA [2]. The distribution of participants with the two criteria in each of the groups is shown in Figure 2.
However, the Framingham study [3], in which the prevalence of HVA was (17.7%), studied 3196 subjects aged 50 years or older (mean age 62±9 years, 56% women), defined AVH as absence of hypertension and pulse wave velocity < 7.6 m/s.
In the MARE study [4] the prevalence of HVA was 9.3% and of EVA 10.4%. This study included 18490 participants with metabolic syndrome and no history of cardiovascular disease (mean age 52±16 years) and defined HVA as a cfPWV value below the age quintile-specific 10th percentile in participants aged < 35 years the lowest 10th percentile of cfPWV distribution.
The Shanghai study [5] EVA prevalence was 30.6%, with 2098 subjects (mean age 71.3±6.1 years, 55.5% women). HVA was defined as the absence of hypertension and a normal carotid-femoral PWV according to age and blood pressure of the participants.
In the OPTIMO study [6] the prevalence of EVA was 5.7% (mean age 49.9±15.5 years, 50% women) and 1416 subjects from 12 Latin American countries were included. In this study cfPWV was measured using an oscillometric device (Mobil-O-Graph) and EVA was defined when the z-score of cfPWV was ≥ 1.96.
Finally, in the study by Cunha PG et al. [7], the overall prevalence of EVA was 12.5%, including 2542 subjects aged between 18 and 96 years (mean age 45.5years, 55% women). The subjects were classified with EVA if their PWV was at least 97.5th percentile of z-score for mean PWV values adjusted for age (using normal European reference values as comparators).
As we can see, the characteristics of the subjects studied in terms of mean age, sex, race, and criteria used in the definition of vascular ageing are different in these studies.
For better understanding we have added the following paragraph to the manuscript
(Page 11, line 300)
However, the prevalence found between the studies are different and cannot be compared, because the criteria used to define HVA and EVA are not the same. Furthermore, the mean age, sex distribution, as well as the prevalence of different cardiovascular risk factors and race differ from one study to another.
- I would like to revise, if possible, the supplementary material, not attached.
Authors' Answer
We have attached a pdf file of supplementary material.
References
- Williams B, Mancia G, Spiering W, Agabiti Rosei E, Azizi M, Burnier M, Clement D, Coca A, De Simone G, Dominiczak A et al: 2018 Practice Guidelines for the management of arterial hypertension of the European Society of Hypertension and the European Society of Cardiology: ESH/ESC Task Force for the Management of Arterial Hypertension. J Hypertens 2018, 36(12):2284-2309.
- Gomez-Sanchez M, Gomez-Sanchez L, Patino-Alonso MC, Cunha PG, Recio-Rodriguez JI, Alonso-Dominguez R, Sanchez-Aguadero N, Rodriguez-Sanchez E, Maderuelo-Fernandez JA, Garcia-Ortiz L et al: Vascular aging and its relationship with lifestyles and other risk factors in the general Spanish population: Early Vascular Ageing Study. J Hypertens 2020, 38(6):1110-1122.
- Niiranen TJ, Lyass A, Larson MG, Hamburg NM, Benjamin EJ, Mitchell GF, Vasan RS: Prevalence, Correlates, and Prognosis of Healthy Vascular Aging in a Western Community-Dwelling Cohort: The Framingham Heart Study. Hypertension 2017, 70(2):267-274.
- Nilsson PM, Laurent S, Cunha PG, Olsen MH, Rietzschel E, Franco OH, RyliškytÄ— L, Strazhesko I, Vlachopoulos C, Chen CH et al: Characteristics of healthy vascular ageing in pooled population-based cohort studies: the global Metabolic syndrome and Artery REsearch Consortium. J Hypertens 2018, 36(12):2340-2349.
- Ji H, Teliewubai J, Lu Y, Xiong J, Yu S, Chi C, Li J, Blacher J, Zhang Y, Xu Y: Vascular aging and preclinical target organ damage in community-dwelling elderly: the Northern Shanghai Study. J Hypertens 2018, 36(6):1391-1398.
- Botto F, Obregon S, Rubinstein F, Scuteri A, Nilsson PM, Kotliar C: Frequency of early vascular aging and associated risk factors among an adult population in Latin America: the OPTIMO study. J Hum Hypertens 2018, 32(3):219-227.
- Cunha PG, Cotter J, Oliveira P, Vila I, Boutouyrie P, Laurent S, Nilsson PM, Scuteri A, Sousa N: Pulse wave velocity distribution in a cohort study: from arterial stiffness to early vascular aging. J Hypertens 2015, 33(7):1438-1445.

Reviewer 2 Report
Well-cited report and study analysis. Needs improvement in grammar.
Author Response
Comments and Suggestions for Authors
Well-cited report and study analysis. Needs improvement in grammar.
Authors' Answer
Following the indications of the review, we have edited the manuscript. A certificate of editing is enclosed.

Reviewer 3 Report
The manuscript entitled „Association of insulin resistance with vascular ageing in general Caucasian population. EVA study.” written by Leticia Gómez-Sánchez et al. presents interesting results that reveal relationships between different indices of insulin resistance and vascular aging measured by cfPWV and baPWV parameters. The results obtained indicate that insulin resistance is an independent predictor of vascular ageing. The reviewed work is properly designed and scientifically significant. Overall, the text is well written and the English language is appropriate and understandable. The manuscript also has some weak points, and my suggestions of improvements are addressed in the comments below.
General concept comments:
- Although detailed information about the study population was provided in other papers (references 33 and 35), some information regarding the participants should be added. Section 2.1 should be extended with information about nationality of the participants (presumably Spanish) and detailed inclusion and exclusion criteria. The authors stated that the study was conducted on subjects without cardiovascular diseases, but there is no information on how the examination of the subject was carried out and what cardiovascular diseases were evaluated. Figure 1 shows that the inclusion criteria were age, sex and provided acceptance, however, the criteria belong to the “other” category are mysterious and need to be explained.
Specific comments:
- I have found some editorial errors in the manuscript. Although such errors will be identified by publisher staff at a later stage of the manuscript processing, I would like to address some points which need to be checked by authors.
- Line 27 – “Results and Conclusions” should be removed
- Line 80 – the beginning of this sentence is missing
- Line 124 – “annd” should be changed to “and”
- Line 209 - “Figure 2…” should be changed to “Figure 3…”
- The legend for Figure 4 includes a sentence in Spanish, please check it
- Line 273 – some words are missing
- Check the alignment of the formatting of the reference list with the journal requirements.
- The font in Figures 1, 2 and 4 is very small, thus these figures should be enlarged and figure legends placed beneath the figures.
- The format of the Figure 3 (barplot) is a little confounding. I propose authors to divide this figure into two separate panels, one for cfPWV and the other for baPWV results. This plot could also be changed to 2D plot. Label for Y axis (e.g. Percentage) should be added.
I believe that my suggestions will be helpful to the authors to increase the quality of the reviewed work.
Author Response
Comments and Suggestions for Authors
The manuscript entitled: “Association of insulin resistance with vascular ageing in general Caucasian population. EVA study.” written by Leticia Gómez-Sánchez et al. presents interesting results that reveal relationships between different indices of insulin resistance and vascular aging measured by cfPWV and baPWV parameters. The results obtained indicate that insulin resistance is an independent predictor of vascular ageing. The reviewed work is properly designed and scientifically significant. Overall, the text is well written and the English language is appropriate and understandable. The manuscript also has some weak points, and my suggestions of improvements are addressed in the comments below.
General concept comments:
- Although detailed information about the study population was provided in other papers (references 33 and 35), some information regarding the participants should be added. Section 2.1 should be extended with information about nationality of the participants (presumably Spanish) and detailed inclusion and exclusion criteria. The authors stated that the study was conducted on subjects without cardiovascular diseases, but there is no information on how the examination of the subject was carried out and what cardiovascular diseases were evaluated. Figure 1 shows that the inclusion criteria were age, sex and provided acceptance, however, the criteria belong to the “other” category are mysterious and need to be explained.
Authors' Answer
Following the reviewer's recommendations, we have expanded the information in 2.1. The cardiovascular diseases assessed were ischaemic heart disease or stroke, peripheral arterial disease, or heart failure. To assess the presence of these diseases and others, we reviewed primary care and hospital care records, as well as a subject survey.
The category "other" corresponds to 74 subjects selected in the sample but could not be located due to change of address or telephone number.
The population base of the present study was the list of subjects receiving health care in five urban Spanish primary care health centres (43,946 persons). From this population all subjects born in 1941 (M: 506, F: 718), 1951 (M: 880, F: 1125), 1961 (M: 1057, F: 1310), 1971 (M: 1028, F: 1274) and 1981 (M: 1091, F: 1197) were listed, separated by sex.
From these lists, 50 women and 50 men in each group were selected by random sampling with replacement, a total of 500 subjects. A total of 510 subjects were discarded for the following reasons: 259 subjects did not meet the inclusion criteria, 177 did not agree to participate in the study, and 74 subjects could not be located due to a change of address or telephone number (Other). Therefore, these 510 subjects, using random sampling with replacement, were replaced in the respective sampling strata.
The current version is as follows (Page, 3 line 76):
The included individuals are from an urban population of 43,946 people. Through random sampling with replacement stratified by sex and age groups (35, 45, 55, 65 and 75 years), 501 Spanish individuals were selected, with approximately 100 in each of the group (50 women and 50 men). The recruitment was carried out between June 2016 and November 2017. Inclusion criteria: Patients aged 35–75 years who agree to participate in the study and do not meet any of the exclusion criteria. Inclusion criteria: patients aged 35–75 years who agreed to participate in the study and did not meet any of the exclusion criteria. Exclusion criteria: participants who are in terminal condition, could not travel to the health centres to undergo the corresponding examinations, and those who did not wish to sign the informed consent; participants with a medical history of CVD (ischaemic heart disease or stroke, peripheral arterial disease or heart failure), diagnosed renal failure in terminal stages (glomerular filtration rate below 30%), chronic inflammatory disease or acute inflammatory process in the past 3 months; and patients treated with oestrogens, testosterone or growth hormones. A detailed description of the reference population, as well as the inclusion and exclusion criteria and the causes by age group and sex is shown in Figure 1.
Figure 1. Flow diagram describing the reference population (43,946), those included and excluded, as well as exclusion criteria by age group and sex. A total of 259 subjects did not meet the inclusion criteria; 177 did not agree to participate in the study; 74 (other) subjects were not located due to changes in address or telephone number.
Specific comments:
- I have found some editorial errors in the manuscript. Although such errors will be identified by publisher staff at a later stage of the manuscript processing, I would like to address some points which need to be checked by authors.
- Line 27 – “Results and Conclusions” should be removed
- Line 80 – the beginning of this sentence is missing
- Line 124 – “annd” should be changed to “and”
- Line 209 - “Figure 2…” should be changed to “Figure 3…”
- The legend for Figure 4 includes a sentence in Spanish, please check it
- Line 273 – some words are missing
- Check the alignment of the formatting of the reference list with the journal requirements.
Authors' Answer
We have edited the manuscript. We attach an edition certificate.
- We have deleted Results and Conclusions in line 27 of the abstract.
- The sentence in the current version reads as follows:
A detailed description of the reference population, as well as the inclusion and exclusion criteria and the causes by age group and sex is shown in Figure 1.
- We have corrected the typographical error in line 124 to read as follows in the current version.
The percentiles by age groups and sex of cfPWV and baPWV are shown in Additional file 1: Fig. S1.
- We have renumbered figure 2 by figure 3.
- We have translated the Spanish words into English. In the current version of the manuscript the legend for Figure 4 appears as:
Figure 4. Odds ratio of determinants of EVA vs. HVA + NVA with insulin resistance indexes in adults ages 35–75 years. Adjusted for age, sex, antihypertensive drugs, lipid-lowering and hypoglycaemic drugs.
Abbreviations: EVA early vascular aging, HVA healthy vascular aging, NVA normal vascular aging, cfPWV carotid to femoral pulse wave velocity, baPWV brachial to ankle pulse wave velocity, HOMA-IR homeostatic model assessment of insulin resistance, TyG index triglyceride and glucose index, T/HDL-c ratio triglyceride to high-density lipoprotein cholesterol ratio, LAP index lipid accumulation product index, WHt ratio waist-to-height ratio, VA index visceral adiposity index.
- The results obtained from the logistic regression analysis using model 2 and model 3 can be seen in Additional file 1. Fig. S3 and Fig. S4.
- We have revised and adapted the references to the format recommended by the journal using the EDNOTE bibliographic manager
- The font in Figures 1, 2 and 4 is very small, thus these figures should be enlarged and figure legends placed beneath the figures.
Authors' Answer
We have increased the font size of the figures and placed the caption at the bottom of the figures.
- The format of the Figure 3 (barplot) is a little confounding. I propose authors to divide this figure into two separate panels, one for cfPWV and the other for baPWV results. This plot could also be changed to 2D plot. Label for Y axis (e.g. Percentage) should be added.
Authors' Answer
We have modified the figure as recommended by the reviewer.
We have labelled the Y axis, and divided Figure 3 into Figure 3a and Figure 3b.
In the manuscript we have kept it in 3 dimensions, as we think the image is better than in two dimensions. However, we have also included the two-dimensional figure in an additional separate file in case it is considered more appropriate in the journal.
I believe that my suggestions will be helpful to the authors to increase the quality of the reviewed work.
I would like to thank you for the suggestions in the revision which have considerably improved the quality of the manuscript.